# About Cogredient and Contragredient Linear Differential Equations †

**Vasily Gorelov**

Moscow Power Engineering Institute, National Research University, 111250 Moscow, Russia; gorelov.va@mail.ru; Tel.: +7-495-362-77-74

† The results of the work are obtained in the framework of the state contract of the Ministry of Education and Science of the Russian Federation (project no. FSWF-2020-0022).

**Abstract:** The notions of cogredience and contragredience, which have great importance to the question of algebraic independence of linear differential equation solutions, are discussed in the paper. Conditions of equivalence of two definitions of cogredience and contragredience are found.

**Keywords:** Siegel's method; algebraic independence; hypergeometric functions; E-functions

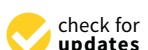

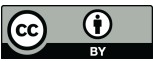

## 1. Introduction

Let $M(q, K)$ be the set of all matrices of size $q \times q$ with elements from a ring $K$, $GL(q, K)$ be the set of all invertible matrices in $M(q, K)$, $\mathbb{C}[z^{\pm 1}]$ be the ring $\mathbb{C}[z, z^{-1}]$, $\mathbb{F}\langle v_1, \ldots, v_n \rangle$ be the smallest differential field containing the field $\mathbb{F}$ and functions $v_1, \ldots, v_n$ (see Chapter 1 in [1]).

In the theory of transcendental numbers, one of the main methods remains the Siegel–Shidlovsky method (see [2,3]), with which one can prove transcendence and algebraic independence of the values of entire functions of a certain class (so-called E-functions).

Siegel calls the entire function

$$f(z) = \sum_{n=0}^{\infty} c_n \frac{z^n}{n!}$$

an E-function, if:

(1)    all the numbers $c_n$ belong to an algebraic field $\mathbb{K}$ of finite degree over $\mathbb{Q}$;

(2)    for arbitrary $\varepsilon > 0$ $\quad \overline{|c_n|} = O(n^{\varepsilon n}), \quad n \to \infty$, where $\overline{|\alpha|}$ is the maximum of the absolute values of the algebraic number $\alpha$ and all its conjugates in the field $\mathbb{K}$;

(3)    for arbitrary $\varepsilon > 0$ the least common denominator of $c_1, \ldots, c_n$ is $O(n^{\varepsilon n}), \quad n \to \infty$.

To apply the Siegel–Shidlovsky method, it is necessary that the functions under consideration constitute a solution of a system of differential equations and be algebraically independent over $\mathbb{C}(z)$.

The question of algebraic independence of solutions of linear differential equations and systems of such equations is also of great importance in differential algebra, analytical theory of differential equations, theory of special functions and calculus in a broad sense. As shown in the works of E. Kolchin [4], F. Beukers, W. Brownwell and G. Heckman [5], this question is largely reduced to checking the cogredience and contragredience conditions.

Two systems of linear homogeneous differential equations of the 1st order,

$$\vec{y}' = A_k \vec{y}, \quad A_k \in M(q, \mathbb{C}(z)), \quad q \geqslant 2, \quad k = 1, 2,$$

are called cogredient (respectively, contragredient) if, for arbitrary fundamental matrices $\Phi_1$, $\Phi_2$ of these systems, one of the equalities

$$\Phi_1 = gB\Phi_2C, \quad \Phi_1(\Phi_2C)^T = gB, \tag{1}$$

holds, where $C \in GL(\mathbb{C})$, $B \in GL(\mathbb{C}(z))$, $g = g(z)$ is a function with the condition $g'/g \in \mathbb{C}(z)$. Similarly, the concepts of cogredience and contragredience are determined for linear homogeneous differential equations of arbitrary order.

**Example 1.** *Consider Kummer's functions*

$$A_{\mu,\nu}(z) = {}_1F_1\left( \begin{array}{c} \nu \\ \mu \end{array} \middle| z \right) = \sum_{n=0}^{\infty} \frac{\nu(\nu+1)\dots(\nu+n-1)}{n!\mu(\mu+1)\dots(\mu+n-1)} z^n,$$

*satisfying the equations*

$$y'' + \left(-1 + \frac{\mu}{z}\right)y' - \frac{\nu}{z}\, y = 0. \tag{2}$$

*If $\mu \notin \mathbb{Z}$, then the collection of functions $\{A_{\mu,\nu}(z),\ z^{1-\mu}A_{2-\mu,\,\nu-\mu+1}(z)\}$ is the fundamental system of solution of Equation (2) (see, for example, Chapter 7 in [6]). Suppose that $\Phi$ is the fundamental matrix corresponding to this collection. Suppose that $\Psi$ is the fundamental matrix, which is determined similarly for the function $A_{\mu,\nu+1}(z)$ and Equation (2), where we change $\nu$ to $\nu + 1$. Then the identity*

$$\Psi = \frac{1}{\nu}\left( \begin{array}{cc} \nu & z \\ \nu & z+\nu-\mu+1 \end{array} \right)\Phi$$

*holds (see [7]).*

　　More complex examples are given at the end of the article.

　　Examples of cogredience can be obtained from the relations of contiguity for the hypergeometric function, discovered by Gauss. For arbitrary generalized hypergeometric functions, the cogredience conditions were obtained by the author (see [8]; Lemma 12 in [7]).

　　Recall that generalized hypergeometric functions (see [2,3,6,9]) are functions of the form

$$_l\varphi_q(z) = {}_l\varphi_q(\vec{\nu};\vec{\lambda};z) = {}_{l+1}F_q\left( \begin{array}{c} 1,\nu_1,\dots,\nu_l \\ \lambda_1,\dots,\lambda_q \end{array} \middle| z \right) = \sum_{n=0}^{\infty} \frac{(\nu_1)_n \dots (\nu_l)_n}{(\lambda_1)_n \dots (\lambda_q)_n} z^n,$$

where $0 \le l \le q$, $(\nu)_0 = 1$, $(\nu)_n = \nu(\nu+1)\dots(\nu+n-1)$, $\vec{\nu} = (\nu_1,\dots,\nu_l) \in \mathbb{C}^l$, $\vec{\lambda} \in (\mathbb{C} \setminus \mathbb{Z}^-)^q$.

　　The function $_l\varphi_q(\vec{\nu};\vec{\lambda};z)$ satisfies (generalized) hypergeometric differential equation

$$L(\vec{\nu};\vec{\lambda};z)y = (\lambda_1-1)\dots(\lambda_q-1),$$

where

$$L(\vec{\nu};\vec{\lambda};z) \equiv \left( \prod_{j=1}^{q}(\delta+\lambda_j-1) - z\prod_{k=1}^{l}(\delta+\nu_k) \right), \quad \delta = z\frac{d}{dz}.$$

　　An explicit form of the equation $L(\vec{\nu};\vec{\lambda};\alpha z^p)y = 0$, obtained from $L(\vec{\nu};\vec{\lambda};z)y = 0$ by the substitution $z \longrightarrow \alpha z^p$, where $\alpha \in \mathbb{C}$, $p \in \mathbb{N}$, is given in [8,10]. The Wronskian of the equation $L(\vec{\nu};\vec{\lambda};\alpha z^p)y = 0$ is Lemma 6 in [10]

$$W = cz^{-(\lambda_1+\dots+\lambda_q-q)p-(q-1)q/2}(1-\alpha z^p)^{(\lambda_1+\dots+\lambda_q-\nu_1-\dots-\nu_q-q)\varepsilon}e^{\alpha z^p\varepsilon_1},$$

where $c \in \mathbb{C} \setminus \{0\}$, $\varepsilon = \delta_q^l$, $\varepsilon_1 = \delta_q^{l+1}$, $\delta_i^j$ is the Kronecker delta.

　　If $\vec{\nu} \in \mathbb{Q}^l$, $\vec{\lambda} \in \mathbb{Q}^q$, $l < q$, $\alpha$ is an algebraic number, then the function $_l\varphi_q(\vec{\nu};\vec{\lambda};\alpha z^{q-l})$ is an E-function (see [2]; Chapter 5 in [3]).

The first example of contragredience (without mentioning this term) of a system of differential equations related to hypergeometric functions was apparently constructed by Yu.V.,Nesterenko (see Lemma 8 in [11]). Cogredience and contragredience theorems for generalized hypergeometric differential equations, many of which had necessary and sufficient conditions, were proved by the author in [8] and [7] (Lemma 14). At the same time, in article [8], in fact a narrower definition was used. According to this definition, we have the equality (1)

$$g = z^r e^{\gamma z} \ \ or \ \ g = z^r \exp(\gamma z^p + \gamma_1 z^{p_1}), \tag{3}$$

where $r, \gamma, \gamma_1 \in \mathbb{C}$, $p, p_1 \in \mathbb{N}$. The examples constructed in [10–12] were also related to case (3).

In this paper, we find the conditions under which the definitions of cogredient and contragredient equations or systems in equalities (1), we can restrict ourselves to case (3).

**Theorem 1.** *Let* $\Phi_k = \|v_{k,t,s}\|_{t,s=1,\dots,q}$ *be the fundamental matrices of the systems*

$$\vec{v}'_k = A_k \vec{v}_k, \ \ A_k \in M(q, \mathbb{C}[z^{\pm 1}]), \ \ q \geq 2, \ \ k = 1, 2,$$

*moreover*

$$\deg tr_{\mathbb{C}(z, W_k)} \mathbb{C}\langle v_{k,1,1}, \dots, v_{k,1,q} \rangle = q^2 - 1,$$

$$W_k = |\Phi_k| = c_k z^{\sigma_k} \exp(\alpha_k z^{p_k}), \ \ p_k \in \mathbb{N}, \ \ c_k, \sigma_k, \alpha_k \in \mathbb{C}, \ \ k = 1, 2.$$

*Then, for the cogredience and contragredience of systems* $(A_1)$, $(A_2)$, *it is necessary and sufficient to satisfy equalities* (1) *with conditions* (3).

## 2. Proof of the Theorem 1

**Lemma 1.** *(Lemma 6 in [8]). Suppose that* $\mathbb{F}$ *is a differential field with field of constants* $\mathbb{C}$. *Suppose that, for any* $k$, $1 \leq k \leq n$ $\Phi_k = \|v_{k,s}^{(i)}\|_{i=0,\dots,q_k-1; s=1,\dots,q_k}$ *is the fundamental matrix of the differential equation*

$$y^{(q_k)} + P_{k,q_k-1} y^{(q_k-1)} + \cdots + P_{k,0} y = 0, \ \ q_k \geq 2, \ \ P_{k,s} \in \mathbb{F},$$

*and* $|\Phi_k| \in \mathbb{F}$. *Suppose that the field of constants of the differential field* $\mathbb{L} = \mathbb{F}\langle v_{1,1}, \dots, v_{n,q_n} \rangle$ *is* $\mathbb{C}$, *and* $\deg tr_{\mathbb{F}} \mathbb{L} < q_1^2 + \cdots + q_n^2 - n$. *Then, either* $\deg tr_{\mathbb{F}} \mathbb{F}\langle v_{k,1}, \dots, v_{k,q_k} \rangle < q_k^2 - 1$ *for some* $k$ *or, for some indices* $1 \leq j < k \leq n$ $q_j = q_k = q$ *holds as well as at least one of the following equalities:*

$$\Phi_j = aB\Phi_k C, \ \ \Phi_j = aB(\Phi_k^{-1})^T C, \tag{4}$$

*where* $a \in \mathbb{L}$, $a^q \in \mathbb{F}$, $B \in GL(q, \mathbb{F})$, $C \in GL(q, \mathbb{C})$.

An analog of Lemma 1 for $q_1 = \cdots = q_n$ was proved by E. Kolchin [4].

For groups $Sp(2k, \mathbb{C})$ and $SO(n, \mathbb{C})$ there is a generalization of the assertion of Lemma 1 (see Proposition 1.8.2 in [13]). An implicit analog of Lemma 1 for Galois groups containing $SL(m), Sp(2k)$, was used in [5] in the proof of the Theorem 2.3.

Assuming in Lemma 1 $\mathbb{F} = \mathbb{C}(z)$ and changing, if necessary, the numbering of equations, equalities (4) can be written in the form

$$\Phi_1 = A\Phi_2 C, \ \ \Phi_1 = A(\Phi_2^{-1})^T C, \tag{5}$$

where $A$ is a matrix whose elements are analytic functions, generally speaking, are ambiguous.

Let $A_k$ be the matrix of coefficients of the system, corresponding to the differential equation with number $k$, $k = 1, \dots, n$.

The next Lemma follows from Lemma 1 (see Lemma 7 in [8])

**Lemma 2.** *If under the conditions of Lemma 1* $\mathbb{F} = \mathbb{C}(z)$, $A_j \in M(q, \mathbb{C}[z^{\pm 1}])$, $j = 1, \dots, n$, *then in equalities* (5) $A = z^r B$, $B \in GL(q, \mathbb{C}[z^{\pm 1}])$, $r \in \mathbb{Q}$, $rq \in \mathbb{Z}$.

**Lemma 3.** *Let $\mathbb{V}$ be an arbitrary differential field of analytic functions containing $\mathbb{C}(z)$ but not containing irrational functions, whose logarithmic derivatives belong to $\mathbb{C}(z)$. Then, any functions linearly independent over $\mathbb{C}(z)$, whose logarithmic derivatives belong to $\mathbb{C}(z)$ will be linearly independent over $\mathbb{V}$.*

**Proof of Lemma 3.** Let

$$\varkappa_1 f_1 + \cdots + \varkappa_n f_n = 0, \tag{6}$$

$$\varkappa_k \in \mathbb{V}, \ \ \varkappa_k \not\equiv 0, \ \ f_k'/f_k = g_k \in \mathbb{C}(z), \ \ k = 1, \ldots, n,$$

where $n \geq 2$ is the smallest possible. Differentiating equality (6), we obtain

$$(\varkappa_1' + \varkappa_1 g_1) f_1 + \cdots + (\varkappa_n' + \varkappa_n g_n) f_n = 0. \tag{7}$$

Since the number $n$ is minimal, the linear combinations on the left-hand sides equalities (6) and (7) must be proportional. In this way,

$$\frac{\varkappa_1'}{\varkappa_1} + g_1 = \frac{\varkappa_k'}{\varkappa_k} + g_k, \ \ k = 1, \ldots, n.$$

Hence, $(\varkappa_k/\varkappa_1)'/(\varkappa_k/\varkappa_1) \in \mathbb{C}(z)$ and, since $\varkappa_k/\varkappa_1 \in \mathbb{V}$, then $\varkappa_k/\varkappa_1 \in \mathbb{C}(z)$, $k = 1, \ldots, n$. $\square$

**Lemma 4.** *Let $\Phi_k = \|v_{k,i,s}\|_{i,s=1,\ldots,q_k}$ be the fundamental matrix of the system*

$$\vec{v}_k' = A_k \vec{v}_k, \ \ A_k \in M(q_k, \mathbb{C}(z)), \ \ q_k \geq 2, \tag{8}$$

$|\Phi_k| = W_k = |v_{k,i,s}|_{i,s} \in \mathbb{C}(z)$, $k = 1, \ldots, n$ *and the functions*

$$\left\{ v_{k,i,s} \big|_{k=1,\ldots,n; \, i,s=1,\ldots,q_k; \, (i,s) \neq (q_k,q_k)} \right\} \tag{9}$$

*be algebraically independent over $\mathbb{C}(z)$. Then the field $\mathbb{V}$ generated over $\mathbb{C}(z)$ by functions (9) does not contain irrational functions whose logarithmic derivatives belong to $\mathbb{C}(z)$.*

**Corollary 1.** *Any linearly (algebraically) independent over $\mathbb{C}(z)$ functions whose logarithmic derivatives belong to $\mathbb{C}(z)$, under the conditions of Lemma 4 will be linearly (respectively, algebraically) independent over $\mathbb{V}$.*

**Proof of Lemma 4.** If functions (9) are algebraically independent over $\mathbb{C}(z)$, then it is convenient to carry out all operations with them formally, as with their corresponding variables

$$\left\{ x_{k,i,s} \big|_{k=1,\ldots,n; \, i,s=1,\ldots,q_k; \, (i,s) \neq (q_k,q_k)} \right\}. \tag{10}$$

The fundamental matrix $\Phi_k$ takes the form

$$\Phi_k = \begin{pmatrix} x_{k,1,1} & \cdots & x_{k,1,q_k} \\ \cdots & \cdots & \cdots \\ x_{k,q_k,1} & \cdots & \hat{x}_{k,q_k,q_k} \end{pmatrix},$$

where $\hat{x}_{k,q_k,q_k}$ is a rational function of variables (10), defined by the equation $|\Phi_k| = b_k \in \mathbb{C}(z)$, equivalent to

$$A_{k,q_k,1} x_{k,q_k,1} + \cdots + A_{k,q_k,q_k-1} x_{k,q_k,q_k-1} + A_{k,q_k,q_k} \hat{x}_{k,q_k,q_k} = b_k,$$

from where

$$\hat{x}_{k,q_k,q_k} = \frac{b_k - A_{k,q_k,1} x_{k,q_k,1} - \cdots - A_{k,q_k,q_k-1} x_{k,q_k,q_k-1}}{A_{k,q_k,q_k}}, \tag{11}$$

where $A_{k,q_k,1}, \ldots, A_{k,q_k,q_k}$ are algebraic complements the corresponding elements of the matrix $\Phi_k$, which are polynomials in variables (10). Note that for a different choice of functions (9) included in $W_k$, the function $b_k \in \mathbb{C}(z)$, generally speaking, is multiplied by some factor from $\mathbb{C}$.

The derivatives with respect to $z$ of variables (10) can be calculated formally, proceeding from the systems of Equation (8) and equalities (11).

Let the function $v$ satisfy the equation

$$y' = ay, \ \ a \in \mathbb{C}(z) \tag{12}$$

and belongs to the field $\mathbb{V}$, that is, it can be represented in the form

$$v = T = \frac{P}{Q}, \tag{13}$$

where $T$ is a rational function over $\mathbb{C}$ of functions (9) and $z$, $P$ and $Q$ are polynomials in the same functions, $(P, Q) = 1$.

Replacing functions (9) in equality (13) by variables (10) and differentiating it with respect to $z$, we obtain

$$v' = \frac{P_1}{Q_1},$$

where $P_1$, $Q_1$ are polynomials in variables (10) and $z$, $(P_1, Q_1) = 1$. In view of equality (12)

$$\frac{P_1}{Q_1} = a\frac{P}{Q}$$

identically by (10) and $z$. This implies, that if in equality (13) instead of $\vec{v}_{k,1}, \ldots, \vec{v}_{k,q_k}$, $k = 1, \ldots, n$ we substitute any other linearly independent solutions of the corresponding systems $(A_k)$, such that $W_k = b_k$, then the function $u = T$ will be a solution of Equation (12) and, therefore, $u = cv$, $c \in \mathbb{C}$. Let $T$ really depend on the variables included in the matrix $\Phi_1$, and $q_1 \geq 3$. Substitute in $T$ instead of variables $x_{1,i,1}$ functions $v_{1,i,1} + \lambda v_{1,i,2}$, $i = 1, \ldots, q_1$, where $\lambda$ is a new variable, and instead of the remaining variables (10), the corresponding functions (9). Obviously, the Jacobian $W_k$ will not change in this case. Then

$$c(\lambda)v = T(v_{1,1,1} + \lambda v_{1,1,2}, \ldots, v_{1,q_1,1} + \lambda v_{1,q_1,2}, v_{1,1,2}, \ldots, v_{n,q_n-1,q_n}). \tag{14}$$

In view of the algebraic independence of functions (9), this equality is preserved when replacing functions (9) with the corresponding variables (10). Differentiating after such a change equality (14) with respect to $\lambda$ and then setting $\lambda = 0$, we get

$$c'(0)T = \frac{\partial T}{\partial x_{1,1,1}}x_{1,1,2} + \frac{\partial T}{\partial x_{1,2,1}}x_{1,2,2} + \cdots + \frac{\partial T}{\partial x_{1,q_1,1}}x_{1,q_1,2}. \tag{15}$$

We define the degree of a rational function with respect to any set of variables as the difference between the degrees of the numerator and denominator for this population. It is easy check that, with such a definition, the degree of the product of rational functions is equal to the sum of the powers of the factors, the degree of the sum does not exceed the maximum degrees of terms, and when taking a partial derivative with respect to some variable from the selected population, the degree decreases. Hence, the degree of the right-hand side of equality (15) with respect to the set of variables $x_{1,1,1}, \ldots, x_{1,q_1,1}$ is strictly less than the degree of the left side, except for the case when $T$ does not depend on these variables, and $c'(0) = 0$. Exactly the same reasoning shows that in the case of $q_1 \geq 3$ $T$ does not depend on $x_{1,1,s}, \ldots, x_{1,q_1,s}$, $2 \leq s \leq q_1$, and in case $q_1 = 2$ $T$ does not depend on $x_{1,1,2}$. It remains to prove that for $q_1 = 2$ $T$ does not depend on $x_{1,1,1}$, $x_{1,2,1}$.

Repeating the previous reasoning taking into account that according to Liouville's formula $v_{1,2,2} = (v_{1,2,1}v_{1,1,2} + b_1)/v_{1,1,1}$, instead of (15) we get

$$c'(0)Tx_{1,1,1} = \frac{\partial T}{\partial x_{1,1,1}}x_{1,1,1}x_{1,1,2} + \frac{\partial T}{\partial x_{1,2,1}}x_{1,2,1}x_{1,1,2} + \frac{\partial T}{\partial x_{1,2,1}}b_1.$$

Comparing the degrees in the set of variables $x_{1,1,1}$, $x_{1,2,1}$ left and right sides of the resulting equality, we conclude that $T$ does not depend on $x_{1,1,1}$ and $x_{1,2,1}$. Thus, $T$ is independent on variables (10) included in $\Phi_1$, and therefore, on all variables (10). □

The Corollary 1 of Lemma 4 is obtained using Lemma 3 and the fact that any product of powers of functions whose logarithmic derivatives belong to $\mathbb{C}(z)$, is a function with the same property.

**Lemma 5.** *Let the system of equations*

$$\vec{v}' = A\vec{v}, \ \ A \in M(q, \mathbb{C}(z)), \ \ q \geq 2, \tag{16}$$

*have no nontrivial solutions containing zero components, and* $\Phi = \|v_{i,s}\|_{i,s=1,\dots,q}$ *be an arbitrary fundamental matrix of this system. Then for any* $t \in \{1,\dots,q\}$ *the matrix* $\Psi = \|u_s^{(i)}\|_{i=0,\dots,q-1;s=1,\dots,q}$, *where* $u_s = v_{t,s}$, *is the fundamental matrix of a differential equation*

$$v^{(q)} + a_{q-1}v^{(q-1)} + \cdots + a_0 v = 0, \ \ q \geq 2, \ \ a_j \in \mathbb{C}(z),$$

*moreover,* $\Psi = \Omega\Phi$, $\Omega \in GL(q, \mathbb{C}(z))$.

**Corollary 2.** *Let, under the conditions of Lemma 5,* $W = |\Phi|$, $W^\circ = |\Psi|$. *Then*

$$\deg tr_{\mathbb{C}(z,W)}\mathbb{C}\langle u_1,\dots,u_q\rangle = \deg tr_{\mathbb{C}(z,W^\circ)}\mathbb{C}\langle v_{1,1},\dots,v_{q,q}\rangle.$$

**Proof of Lemma 5.** From Lemma 7 of Chapter 3 in [3] it follows that if

$$R_1 = P_{1,1}y_1 + \cdots + P_{1,q}y_q, \ \ P_{1,1},\dots,P_{1,q} \in \mathbb{C}[z], \ \ R_1 \not\equiv 0,$$

and the rank of the set of linear forms

$$R_k = T\frac{d}{dz}R_{k-1}, \ \ k = 2,3,\dots, \ \ T \in \mathbb{C}[z], \ \ TA \in M(q, \mathbb{C}[z]),$$

$$R_k = P_{k,1}y_1 + \cdots + P_{k,q}y_q, \ \ P_{k,i} \in \mathbb{C}[z], \ \ i = 1,\dots,q,$$

is less than $q$, then there is at least one nontrivial solution of system (16), under whose substitution all linear forms $R_k$, $k \geq 1$ vanish. If $R_1 = v_{t,s}$, where $t,s \in \{1,\dots,q\}$, this means that $v_{t,s} \equiv 0$. Then, in view of the conditions of the Lemma, we obtain that for any fixed $s$ and $t$ the functions $v_{t,s},\dots,v_{t,s}^{(q-1)}$ are linearly independent linear forms of $v_{1,s},\dots,v_{q,s}$. The coefficients of these linear forms are independent of $s$ and therefore constitute the matrix $\Omega \in GL(q, \mathbb{C}(z))$, and $\Psi = \Omega\Phi$. □

**Proof of Theorem 1.** It is enough to show what if equalities (1) are not satisfied with conditions (3), then they are not satisfied with the condition $g'/g \in \mathbb{C}(z)$.

It is easy to check that the matrix $\Phi_k^\circ = W_k^{-1/q}\Phi_k$ will be the fundamental matrix of the system $(B_k)$, where

$$B_k = A_k + g_k E, \ \ g_k = -\frac{W_k'}{qW_k} = -\frac{\mathrm{Tr}\,A_k}{q} \in \mathbb{C}[z^{\pm 1}], \ \ |\Phi_k^\circ| = 1, \ \ k = 1,2.$$

The matrix $\Phi_k^\circ$ satisfies the conditions of Lemma 5, therefore the matrix $\Psi_k = \|(W_k^{-1/q}v_{k,1,s})^{(i)}\|_{i=0,\ldots,q-1;\,s=1,\ldots,q}$ is the fundamental matrix of the differential equation

$$v^{(q)} + a_{k,q-1}v^{(q-1)} + \cdots + a_{k,0}v = 0, \quad a_{k,j} \in \mathbb{C}(z),$$

moreover, $\Psi_k = \Omega_k\Phi_k^\circ$, $\Omega_k \in GL(q,\mathbb{C}(z))$, $k = 1, 2$. If equalities (1) with conditions (3) are not satisfied for the matrices $\Phi_k$ for $B \in GL(q,\mathbb{C}[z^{\pm 1}])$, then, according to Lemma 2, they also fail for $\Phi_k^\circ$ for $B \in GL(q,\mathbb{C}(z))$. Then, conditions (4) of Lemma 1, in which we put $\mathbb{F} = \mathbb{C}(z)$, cannot hold for matrices $\Psi_k$. Therefore, from Lemma 1 we obtain that $2q^2 - 2$ of functions

$$\left\{ (W_k^{-1/q}v_{k,1,s})^{(i)}\Big|_{k=1,2;\,i=0,\ldots,q-1;\,s=1,\ldots,q;\,(i,s)\neq(q-1,q)} \right\}$$

are algebraically independent over $\mathbb{C}(z)$. Then, in view of Corollary 2, are algebraically independent over $\mathbb{C}(z)$ $2q^2 - 2$ functions

$$\left\{ W_k^{-1/q}v_{k,t,s}\Big|_{k=1,2;\,t,s=1,\ldots,q;\,(t,s)\neq(q,q)} \right\}.$$

Since $(W_k^{1/q})'/W_k^{1/q} \in \mathbb{C}(z)$, hence, taking into account Corollary 1, we conclude that equalities (1) are impossible. $\square$

## 3. Conclusions

1. Consider the functions

$$K_\lambda(z) = {}_0F_1\left(\begin{array}{c|c} - \\ \lambda+1 \end{array} -\frac{z^2}{4}\right) = \sum_{n=0}^{\infty} \frac{(-1)^n}{n!(\lambda+1)_n}\left(\frac{z}{2}\right)^{2n},$$

which differ from the Bessel's functions $J_\lambda(z)$ with the index $\lambda$ only by multiplier $(z/2)^\lambda(\Gamma(\lambda+1))^{-1}$ and satisfying the equations

$$y'' + \frac{2\lambda+1}{z}y' + y = 0. \tag{17}$$

Theorem 1 allows us, in particular, to describe all algebraic identities between the functions $K_\lambda(z)$ and Kummer's functions $A_{\mu,\nu}(z)$. Consider

**Example 2.** *Suppose that $2\lambda \in \mathbb{C}\setminus\mathbb{Z}$, $\alpha \in \mathbb{C}$, $p \in \mathbb{N}$, and $\Phi_1,\Phi_2,\Phi_3$ are the fundamental matrices corresponding to the collections of functions*
*$\{K_\lambda(\alpha z^p), z^{-2p\lambda}K_{-\lambda}(\alpha z^p)\}$, $\{A_{2\lambda+1,\lambda+1/2}(2i\alpha z^p), z^{-2p\lambda}A_{1-2\lambda,1/2-\lambda}(2i\alpha z^p)\}$,*
*$\{A_{1-2\lambda,1/2-\lambda}(-2i\alpha z^p), -z^{2p\lambda}A_{2\lambda+1,\lambda+1/2}(-2i\alpha z^p)\}$,*
*respectively. $\Phi_1,\Phi_2,\Phi_3$ are fundamental matrices of the 2nd order linear differential equations, which are easy to obtain from Equations (2) and (17). Then the identities*

$$\Phi_1 = e^{-i\alpha z^p}\begin{pmatrix} 1 & 0 \\ -ip\alpha z^{p-1} & 1 \end{pmatrix}\Phi_2,$$

$$\Phi_1\Phi_3^T = 2p\lambda z^{-1}e^{-i\alpha z^p}\begin{pmatrix} 0 & -1 \\ 1 & -ip\alpha z^{p-1} + 2p\lambda z^{-1} \end{pmatrix}$$

*hold (see [12]).*

Are there other algebraic identities between $K_\lambda(z)$ and $A_{\mu,\nu}(z)$?

Theorem 1 is applicable to the functions ${}_l\varphi_q(\vec{v};\vec{\lambda};\alpha z^p)$ for $l < q$. Therefore, the necessary and sufficient conditions of cogredience and contragredience of generalized hypergeometric equations from article [8] that were found for the case (3) are also valid for the general definition (1). This comment also applies to the article [10], where conditions of cogredience

and contragredience were also discussed. According to [8,10], Example 2 is the only case of cogredience and contragredience between the equations that are obtained from (2) and (17) by the substitution $z \longrightarrow \alpha z^p$. Therefore, according to [5,8], other algebraic identities between $K_\lambda(z)$ and $A_{\mu,\nu}(z)$, different from the identities derived from Example 2 do not exist.

2. The lack of cogredience and contragredience allows one to conclude about the algebraic independence of generalized hypergeometric functions over $\mathbb{C}(z)$ (see [5,8]). It follows the algebraic independence of their values (see [2,3]). Using the theorems of Chapters 11–13 of the book [3] (or their more exact analogs from [14]), one can also obtain lower estimates of moduli of polynomials of the values.

**Funding:** This research received no external funding.

**Conflicts of Interest:** The author declare no conflict of interest.

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
