# Peer review of "About Cogredient and Contragredient Linear Differential Equations†"

_axioms, doi:10.3390/axioms10020117_

Round 1
Reviewer 1 Report
The article is a continuation of the author's works on the problem of the so-called cogredience and contragredience of linear differential equations and systems of such equations related to hypergeometric functions.
The article is written interestingly. Most proofs are presented with sufficiently complete strict and detailed considerations.
I would recommend that the author "give some examples of the application of Theorem 1." This will make the paper longer but it will be much more readable and thus much better.
I will gladly look at the updated version.
Reviewer 2 Report
Dear Author,
The paper seems to be sound and potentially interesting but before publication you have to make it accessible to a broader public. Otherwise you will have no chance to be read and recognized. Looking at your profile in Scopus fully supports my first impression get from the view of your paper: you have 11 papers in that standard database but only 1 (one) citation (excluding auto-citations).
First of all, you have to better present the subject of your study. Please be more explicit, please use more general terms, please give some simple and broadly understood examples.
Second, please find (or make) some links of your research to many fields of mathematics where linear systems and plehora of transformations between them are ubiquitos. For instance, gauge trnsformations in theoretical physics and various transformations in the intergable systems theory, see for instance:
1. Isomonodromic, Schlesinger, transformations, e.g.:
"Monodromy preserving deformation of linear ordinary differential equations with rational coefficients I, II" Physica D 2 (1981).
2. Darboux matrices (including polynomial ("Neugebauer's")case, reminding your systems very much):
"Algebraic construction of the Darboux matrix revisited" J. Phys. A 42 (2009).
3. Transfer functions matrix and associated problems in the control system theory (this also looks very close to your research), see, e.g.,
"Factorization problems and operator identities" Usp. Mat. Nauk. 41 (1986)
Finally, I will list specific remarks and comments starting from the beginning ot the text:
line 7: what do you mean by "in M(q,K)"? Just a subset of non-degenerate matrices, I think?
line 7: why to define here C[z,z-1] while using afterwards only C[z]?
lines 7-8: please be much more specific. What do you mean by "obtained by joining"? what do you mean by "differential variables"? something like derivatives with respect to different variables? It would be very interesting, but is hardly seen from your text...
lines 9-18: for me some simple example explaining the essence of this applications would be very welcome
lines 18-21: decidedly too concise. After making some computations I see that this is quite standard (but very important and widely used) problem, connected to gauge transformations, Darboux transformation and transfer function matrices. Please, at least, write explicitly both linear equations, defined by two matrices and (my suggestion) look at the transformation between these to matrices - gauge equivalence in the first case, and gauge equivalence between equation and inverse transposed equation (so, contragradient, indeed), in the second case.
lines 27-28: do you mean "is an E-function" by definition? if not - please give the definition.
Theorem 1: add "(k=1,2)" and change "matrix -> matrices"
Lines: 77-84. Conclusions have to be much more extended, clear and without cryptic references to some places in your earlier papers. Please imagine that somebody would like to see in few minutes the essence of your work.
After revising your submission along lines indicated the paper would be suitable for publication.
Good luck!
Round 2
Reviewer 1 Report
This paper is now ready for publication in Axioms.
Reviewer 2 Report
The revision is not satisfactory. I can repeat most of my remarks almost without changes. The Author assumes that a Reader knows all literature and indicating the source is the only explanation needed. This is far from being true.
"The paper seems to be sound and potentially interesting but before publication you have to make it accessible to a broader public. Otherwise you will have no chance to be read and recognized. Looking at your profile in Scopus fully supports my first impression get from the view of your paper: you have 11 papers in that standard database but only 1 (one) citation (excluding auto-citations)". This remark is even more true than before. Your answer is the best proof.
"First of all, you have to better present the subject of your study. Please be more explicit, please use more general terms, please give some simple and broadly understood examples". The example you gave is not simple enough. I would expect here some elementary equations, easy to verify. Is it possible to choose one equation with constant coefficients, for instance?
"Second, please find (or make) some links of your research to many fields of mathematics where linear systems and plehora of transformations between them are ubiquitos. For instance, gauge trnsformations in theoretical physics and various transformations in the intergable systems theory". Now, this is a secondary problem. First, the Author should present his problem in broader context and simpler terms.
lines 7-8: please be much more specific. What do you mean by "obtained by joining"? what do you mean by "differential variables"? something like derivatives with respect to different variables? It would be very interesting, but is hardly seen from your text...
lines 18-21: decidedly too concise. After making some computations I see that this is quite standard (but very important and widely used) problem, connected to gauge transformations, Darboux transformation and transfer function matrices. Please, at least, write explicitly both linear equations, defined by two matrices and (my suggestion) look at the transformation between these to matrices - gauge equivalence in the first case, and gauge equivalence between equation and inverse transposed equation (so, contragradient, indeed), in the second case.
lines 27-28: do you mean "is an E-function" by definition? if not - please give the definition.
Lines: 77-84. Conclusions have to be much more extended, clear and without cryptic references to some places in your earlier papers. Please imagine that somebody would like to see in few minutes the essence of your work.
After revising your submission along lines indicated the paper would be suitable for publication.
Round 3
Reviewer 2 Report
I appreciate attempts of the Author to improve the paper. The revision is quite satisfactory.
Two points to be considered:
- The symbol Φ1 on the page 2 has two different meanings (index or shift). Please correct, or (at least) add something like "not to be confused with...".
- I still suggest to write down the "two systems of linear homogeneous differential equations" explicitly in the matrix form and to compute counterparts of eqs. (1) in terms of the matrices of these equations. Although the Author do not see much analogies at his moment but, I think, that this form will give some food for thought for many Readers.
